# Comparison of the Physical, Physiological, and Psychological Responses of the High-Intensity Interval (HIIT) and Small-Sided Games (SSG) Training Programs in Young Elite Soccer Players

**DOI:** 10.3390/ijerph192113807

**Published:** 2022-10-24

**Authors:** Zied Ouertatani, Okba Selmi, Santo Marsigliante, Bilel Aydi, Nadhir Hammami, Antonella Muscella

**Affiliations:** 1High Institute of Sports and Physical Education of Kef, University of Jendouba, Kef 7100, Tunisia; 2Department of Biological and Environmental Sciences and Technologies (Di.S.Te.B.A.), University of Salento, 73100 Lecce, Italy

**Keywords:** soccer, aerobic fitness, motivation, enjoyment, mood, training load, well-being, HIIT, SSG physical fitness

## Abstract

We investigated the effects of high-intensity interval (HIIT) and small-sided games (SSG) training programs on physiological and psychological responses in young soccer players. Twenty-four male soccer players (age 16.7 ± 0.9 years) were divided into two groups (HIIT and SSG) and completed a 6-week training programs consisting of two training sessions a week. HIIT consisted of intermittent runs lasting 15 s at the 110% maximum aerobic speed followed by a passive recovery lasting 15 s; SSG was instead structured in a 4 versus 4 players games on a playing field of 25 × 35 m. The muscular power of the lower body was assessed before and after each training session using the 5-jump test to leg length, and two incremental field tests (VAMEVAL test and modified agility *t*-test). Our results show that HIIT and SSG have similar beneficial effects on the variables connected to the soccer-specific performance and the endurance with little influence on neuromuscular performances. Psychological responses were assessed using the “physical activity enjoyment scale”, the Hooper index and the POMS variables. Results showed that SSG offers greater perceived enjoyment than what can be achieved with HIIT. In conclusion, SSG can be considered an effective, more motivating, and fun soccer-specific training that brings about physiological responses and neuromuscular adjustments in young soccer players.

## 1. Introduction

Soccer is a sport characterized by high intensity activities, such as sprinting, and high intensity run, turn and jump; therefore, the training should replicate the physical demands of competitive game [1]. Hence, soccer players need a high fitness level to cope with the physical demands of the game. Most coaches use off-the-ball training drills to develop soccer player fitness [2,3]. High-intensity interval training (HIIT) induces significant adaptations in the skeletal muscles resulting in an increased energy production. This effect is achieved by increases in the activity of crucial mitochondrial enzymes [4]. Moreover, HIIT elicits VO_2max_ enhancement, increases running economy, lower blood lactate accumulation during submaximal exercise, and it may well contribute to improve exercise performance [1]. Recently, scientific research has shown that physical capacity can be maintained or developed using HIIT in the form of soccer-specific exercises, such as small-sided games (SSG) [5,6,7,8], which is practiced on small fields with a small number of players. These exercises are generally used to simultaneously solicit the physical, physiological, technical, and tactical aspects required during the match [5,6,9]. Thus, SSG is very widely used throughout the world, and appreciated by players and coaches, because it solicits different aspects of soccer performance simultaneously [10]; indeed, the use of the ball imposes a specific activity leading to the improvement of several factors related to the performance of a football match [11,12]. In addition, SSG induces more motivation and player pleasure [13,14,15]. Also, this type of exercise improves aerobic capacity [16], VO_2max_ of the player [17,18] and makes it possible to reach a heart rate (HRmax) equivalent to that observed during a match (intensities around 80 to 90% of the maximum heart rate, HRmax) [7,19]. During SSG, the intensity of effort is like that recorded during HIIT [14].

However, several factors, such as changes in the playing field size, the number of players, the verbal encouragement of the technical staff, touch number, game instructions, the number and duration of the series, the total duration of the session, the presence or absence of goalkeepers and availability of balls, would directly impact the effort intensity and the player’s physiological response to SSG [14,20,21].

Interestingly, to our knowledge, only a few studies [22,23] have examined the effects of HIIT and SSG methods on the physical fitness and the level of soccer-specific technical skills in young soccer players (U-16) and none from a motivational point of view.

In particular, the results of a meta-analysis of the relevant scientific literature suggested that HIIT or SSG, also in young soccer, produced similar positive effects in the development of aerobic capacity, power, speed, endurance at speed and football specific performance, but little influence on neuromuscular performance [22]. Additionally, some studies suggested that the competition that occurs in the SSG motivates and facilitates players to reach levels of high-intensity exercise [22]. In fact, many soccer players do not reach their performance potential not just because they lack during training the simulations necessary for the harmonious development of their aerobic capacity, but also for the lack of state of positive and motivated mood.

Therefore, it would be important to determine which of the two training programs, HIIT or SSG, produces the strongest motivational responses in young players, taking into account that structuring situations that are intrinsically interesting, challenging and amusing could develop and enhance intrinsic motivation.

Thus, the present study attempted to compare not only the physical and the physiological adaptations but also the psychological responses of the two training programs in young soccer players.

## 2. Materials and Methods

### 2.1. Participants

Twenty-four soccer players were randomly assigned to one of two training groups. All participants were members of the same team in the national football league with training experiences of 6.1 ± 1.31 years and train an average of 5 sessions per week with one match.

We performed a sample size calculation prior recruitment by using the G*Power software (Version 3.1.9.4, University of Kiel, Kiel, Germany). The analysis revealed that 24 subjects would be sufficient to find significant differences with an 85.88% (actual power) chance of correctly rejecting the null hypothesis.

Subjects were non-smokers and non-drinkers. Those suffering from any acute or chronic disease, or under medical treatment, or injured were excluded from the study. These subjects were divided into two groups based on their playing position (central defender, full-back, attacking midfielder, attacking midfielder and forward). Goalkeepers were excluded from the study because they did not follow the same physical training program. 

Regarding the assignment to the HIIT group (n = 12) or to the SSG group (n = 12) first, all soccer players were divided into two groups (n = 12 each) based on their playing position (central defender, full-back, attacking midfielder, attacking midfielder and forward). Then, they were randomly assigned to each group, SSG or HIIT, by flipping a coin [24]. The characteristics of the subjects are presented in Table 1.

Consistently with a random assignment of two groups, no significant differences between groups were found in mean values for age, height, body mass, and % fat, at baseline (*p* > 0.05 by independent *t*-test).

All subjects, or their parents, signed consent to participate freely and voluntarily in this experiment. This study has been approved by the Human Research Ethics Committee of the “Sports Performance and Physical Rehabilitation, ISSEP du Kef, University of Jendouba, Tunisia” Research Unit and is compatible with institutional ethical requirements for human experimentation, in accordance with the Declaration of Helsinki.

### 2.2. Study Design

The study was conducted during the 2021–2022 football season, in Tunisia. It started after the pre-season preparation, on 19 October and ended on 29 December, for a duration of 6 weeks.

The training interventions were administered as an extension of the regular training; two sessions/week of HIIT or SSG training were performed after a standardized warm-up, for 25 min and were strictly controlled as will be described later (in Section 2.3).

As coaches can psychologically influence the players, they constantly verbally encourage players from both groups, also giving instructions to improve the training/game using football specific terminology.

Before and after 6 weeks of training, all participants completed the evaluations of anthropometric parameters, the VAMEVAL test, the 5 jumps test (5JT) and the Hooper Index and POMS questionnaires. 

For the follow-up we used a training diary in which the participants recorded twice a week (when they performed the HIIT or SSG training sessions), the indices of well-being (the quality of sleep, stress, the level of fatigue and the delayed-onset muscular soreness (DOMS)) [25], the quality of total recovery (TQR) [26], (the working time (in min), the Rated Perceived Exertion (RPE) [27].

In relation to the elements collected (duration of work (in min) and RPE), data processing was carried out to determine the training load, monotony and stress on a daily, weekly and periodic basis.

The measurements were administered by the experimenters themselves. The questionnaires were administered by the interviewer himself. Each player was briefed on how to complete the tests. The questionnaires were delivered to subjects directly in the same place and filled them out on his own. Each measurement was made at the same time of day for each player to avoid circadian variations in the measured variables [28] and under the same conditions. Players were asked not to engage in strenuous activities the day before the measurements and to maintain a consistent workout and sleep routine throughout the study. Table 2 summarizes the 6-week training program (for details see later).

### 2.3. Small-Sided Games (SSG)

Four vs. 4 SSG was performed, as previously [8,29], on an outdoor field with natural grass and pitch size of 25 m × 35 m. The SSG duration was strictly controlled (4 bouts of 4 min duration with 3 min of passive recovery in-between) as reported in other investigations [14,18]. The players were asked to perform at maximum effort during the games and to maintain possession of the ball for the longest possible time. During the SSGs, two coaches were around the pitch to provide new balls when necessary to allow continuity of play during the sessions. All SSGs were played without a goalkeeper.

During the training sessions of the SSG, the players freely occupied the entire space of the pitch.

### 2.4. High-Intensity Interval Training (HIIT)

HIIT was performed on an outdoor field with natural grass. Players covered a predetermined distance in 15-s intervals [3]. After each interval, players passively rested for 15 s, and then began the next 15-s interval but ran in the opposite direction [3]. The distance was individualized according to the maximal aerobic speed (MAS) of each player and corresponded to 110% of their MAS [30]. This task was repeated for 4 bouts of 4 min with 3 min of passive rest between bouts.

### 2.5. Measurements of Exercise Intensity

Heart rate (HR) was measured every 5 s during training sessions and during the Vameval test (for Vameval-HRmax), via a Polar Team 124 Sport System. Heart rate data during training were expressed as a percentage of HRmax and grouped into three training intensities: <80%, 80–90% and >90% [13,30].

The reliability of the Vameval protocol for detecting maximum HR has been previously demonstrated (a Cronbach value α 127 of 0.83) [31].

All players indicated their rating of Rated Perceived Exertion (RPE) (Borg’s CR-10 scale, [31]) and duration of each training, at the end of each session of HIIT and SSG.

The training load, monotony and strain of each player were estimated according to the s-RPE method [27,32].

Training duration was multiplied by the session’s RPE to get each player’s training load for each session (i.e., duration × intensity) [33]. Weekly TL was outened adding training load session each session (TLs). Strain for each week is obtained from the product of total weekly TL and monotony [29,33,34].

### 2.6. Hooper Index

The Hooper Index is the sum of the four subjective ratings: sleep (referring to the night before the assessment), fatigue, stress and delayed onset muscle pain (DOMS) evaluated on a scale of 1 to 7 [25]. The HI is obtained from the sum of the four scores: sleep fatigue, stress, and DOMS. 

### 2.7. Total Quality of Recovery (TQR)

The TQR method proposed by Kenttä and Hassmén [26] was used to assess the quality of recovery in participant. 15 min before each session of HIIT or SSG (two sessions/week), each player was asked to record his perception of the recovery state for the previous 24 h. TQR was measured on the Likert scale ranging from 1 (Very, very poor recovery) to 10 (Very, Very good recovery) [26]. The cronbach’s value of 0.90 in the present study.

### 2.8. Physical Activity Enjoyment Scale (PACES)

PACES is an 18-item questionnaire that evaluates pleasure from physical activity [35]. The participants commented on “how do you feel now about the physical activity you have done” by measuring it with a 7-point scale, from 1 (I like it) to 7 (I hate it). Eleven items reported negative feelings, instead seven were positive. Adding up the elements, you get a score between 18 and 126. Obviously, higher scores mean higher enjoyment. The PACES has reported reliability in children [36] and adults [37].

### 2.9. The Profile of Mood States (POMS)

Profile of mood state (POMS) is a psychological rating scale used to assess transient, distinct mood states. POMS is composed of 65 questions, that provides measures of tension, depression, anger, vigour, fatigue, confusion, and total mood disturbance (TMD) [38]. This questionnaire is scored on a five-point Likert-type scale (0 = not at all, 1 = a little, 2 = moderately, 3 = a lot, 4 = extreme). TMD is obtained by adding the five negative mood subscales (tension, depression, anger, fatigue, confusion) and subtracting vigor [38]. This questionnaire was used to monitor mood state and performance of athletes in team sports [39]. The Cronbach’s ranging from 0.85 to 0.92 in the present study.

### 2.10. Physical Performance

Assessments included the 5-Jump Test (5JT), Vameval test, followed by Modified agility *t*-test. Participants became familiar with TL testing protocols and measurements and during the tests, the researchers were blinded to the study protocol, to minimize the risk of bias. Tests were performed before the study and after the 6-week period. The tests were distributed on two consecutive days: on the first day the players performed the Modified agility *t*-test and after 10 min of rest, they performed the 5-Jump Test (5JT); on the second day the players performed the Vameval test.

To minimize the effects of external variables prior to the test, participants observed a 48-h rest period, diet, and sleep period similar for all.

#### 2.10.1. 5-Jump Test

The 5-Jump Test (5JT) was used to evaluate athletes’ lower limb explosive power and coordination level. The 5JT was performed in outdoor, on natural grass, as previously reported by Chamari et al. [40]. The absolute distance of the 5JT (m) was measured with a tape [40].

#### 2.10.2. Vameval Test

The Vameval is a cardiorespiratory fitness test, performed to obtain the maximal aerobic speed and HRmax [41]. The players were required to run in a 200 m running track, with cones placed every 20-m. The players run for as long as possible on a 200-m track calibrated by ten cones placed every 20 m. The test started at 8 km·h^−1^ and continued with increments of 0.5 km·h^−1^ for every minute. 

The players followed the correct running pace by listening to an audio recording, so that they were in line with each of the placed cones, when the beep sounded. The examined subject ends the test when he is no longer able to maintain the running speed set by the acoustic signal for 2 consecutive times or when he believes that he cannot complete the test. The maximal aerobic speed (MAS, km·h^−1^) was determined by the final velocity reached in the last running lap.

#### 2.10.3. Modified Agility *t*-Test

The modified agility *t*-test has previously been reported to be reliable to assess agility and a predictor of football performance [42]. The test involves 4 changes of direction with 3 modes of movement: forward, sideways, and backwards. Four cones were used to mark the start/finish line (point A), the middle (point B) and the end-points (points C and D) (Figure 1).

Each player was required to start with both feet behind the starting line (point A) and following the sound signal, he must sprint forward 5-m and touch the cone (point B) with the right hand and then move 2.5-m left to another cone (point C) and touch it with the left hand. Then the player must move in speed to the right up to the cone “D” and touches its base with the right hand, and shuffle back to cone “B” with the left hand, after which the athlete runs backwards for 5-m to the finish line (point A). Agility times (s) were recorded using timing gates (Microgate, Bolzano, Italy). Each player performed 3 trials, with the best time used for subsequent analysis.

Before test, participants completed a 10-min warm-up period, including sprinting, lateral shifting, jumping, and dynamic stretching. Every 10 min, 4 players were called by the coach to perform the tests, while the other players were waiting for their turn near the field. Each player performed 3 trials alternating with 3 min of passive recovery, to ensure adequate recovery [42].

### 2.11. Statistical Analyses

All statistical analyses were performed using SPSS statistical analysis software (SPSS version 20.0 for Windows, SPSS Inc., Chicago, IL, USA). Data are expressed as mean ± standard deviation. The hypothesis of normality and the homogeneity of the variance were verified using the Kolmogorov-Smirnov. Student’s paired *t*-tests were used to examine the differences between before and after the training program.

To determine the extent of the differences between before and after the training period, the Hopkins threshold was used to calculate the effect size [43]; and was defined as: <0.2 = trivial, 0.2 to 0.6 = small, >0.6 to 1.2 = moderate, >1.2 to 2.0 = large, >2.0 to 4.0 = very large, and >4.0, almost perfect. The significance level was considered at *p* ≤ 0.05.

To evaluate the effect of “Group” (SSG and HIIT), “Time” (pre and post training programs) and “Interaction” (Group × Time) on physical performance (5-Jump Test, modified agility *t*-test, HRmax and MAS) and mood responses (POMS scores), two-way analysis of variance (ANOVA) was used. The analysis of the Bonferroni post-hoc test was completed when the significant effect of the interaction was found. The results were considered significant at the 95% confidence level (*p* ≤ 0.05).

## 3. Results

### 3.1. Training Load, Training Monotony, and Training Strain Table 3 Data Training Load, Training Monotony, and Training strain during Six Weeks of High Intensity Interval (HIIT) or Small-Sided Games (SSG) Training

Table 3 shows the workloads in the different weeks for the two different types of training. Obviously, as indicated in the materials and methods, there are differences between the weeks, but there are no differences in training load between the two groups (HIIT or SSG, Table 3).

**Table 3 ijerph-19-13807-t003:** Data Training Load, Training Monotony, and Training strain during six weeks of high intensity interval (HIIT) or small-sided games (SSG) training.

Weeks	Method	Load (RPE × Duration)	Monotony	Strain
**1°**	**HIIT**	1540.01 ± 100.12	0.81 ± 0.04	1247.4 ± 81.19
	**SSG**	1520 ± 114.34	0.83 ± 0.05	1261.6 ± 94.7
**2°**	**HIIT**	1843.33 ± 133.8 *	0.87 ± 0.06	1604.69 ± 116.38 **
	**SSG**	1873.77 ± 136.63 **	0.84 ± 0.05	1569.92 ± 114.47 *
**3°**	**HIIT**	1976.66 ± 112.85 *	1.01 ± 0.11	1996.42 ± 113.96 *
	**SSG**	1940.76 ± 121.59 *	0.99 ± 0.09	1920.37 ± 120.21 *
**4°**	**HIIT**	2113.33 ± 168.1 *	1.08 ± 0.11	2282.39 ± 181.53 *
	**SSG**	2099 ± 129.82 *	1.06 ± 0.09	2225.75 ± 137.45 *
**5°**	**HIIT**	1686.78 ± 130.95 ^§§^	0.79 ± 0.04	1331.02 ± 103.35 ^§§§^
	**SSG**	1663.33 ± 122.78 ^§§^	0.75 ± 0.03	1247.49 ± 92.08 ^§§§^
**6°**	**HIIT**	1896.66 ± 138.47 *	0.89 ± 0.06	1688.02 ± 123.18 *
	**SSG**	1920 ± 122.78 *	0.92 ± 0.07	1766.4 ± 112.94 **

* *p* < 0.05; ** *p* < 0.01; ^§§^ *p* < 0.01; ^§§§^ *p* <0.001.

A significant (*p* < 0.05) effect of time was found for training loads, monotony, and strain, in both groups. In fact, training loads, monotonies, and strains volumes increased until reaching a maximum value for the 4th week during the 6-week period (*p* < 0.01, *p* < 0.05, *p* < 0.01, respectively).

Furthermore, the results did not show significant differences between the two groups even for subjective responses (Table 3).

### 3.2. Physiological Variables Measurement

Physicians, psychologists, physical educators, and physical therapists have been trying to reach diagnosis solutions related to overtraining [25,44]. Among the discussed criteria, we can highlight analysis of physiological variables and use of psychological measures to follow the perceptions and emotions of the athletes. We here used the Hooper index (Figure 2), which includes four subjective assessments: sleep, stress, delayed-onset muscle soreness (DOMS), and fatigue. There were no significant differences between the HIIT and SSG groups (*p* > 0.05).

Players’ distinct mood states were assessed by the Profile of Mood States (POMS). POMS scores were measured before and after the 6-week of HIIT or SSG training programs. Among POMS variables, bidirectional analysis of variance with repeated measures (group × time) did not reveal any significant effect of time, group and their interaction on anxiety, anger, confusion, and fatigue scores. Thus, these variables were not influenced by the training period or training methods (Table 4). Instead, during the 6-week of HIIT training program, the TMD score increased significantly, and the vigour score decreased significantly (Figure 3). In fact, a significant effect of group-to-group interaction × time on vigour (group: *p* < 0.001, η^2^ = 0.69; interaction; *p* < 0.001, η^2^ = 0.70, by ANOVA) was observed. The results also revealed a significant effect of group, time, and their interaction on TMD (group: *p* < 0.01, η^2^ = 0.56; time: *p* < 0.05, η^2^ = 0.35; interaction, *p* < 0.05, η^2^ = 0.48, Table 4). 

Regarding the overall enjoyment of physical activity, the PACES score showed a highly significant difference each week between the HIIT and SSG groups.

Specifically, the PACES total score (for each of 6 weeks) in the HIIT group was 59.30 ± 7.17 and was significantly lower than that of the SSG, 88.70 ± 6.99 (*p* < 0.001, η^2^ = 4.44; Figure 4).

### 3.3. Physical Performance Assessement

Before and after each training session the lower body muscular power was assessed using the 5-jump test (5JT) to leg length, and two incremental field tests: VAMEVAL test and modified agility *t*-test.

To have a clear picture of the fitness status of each athlete practicing the two types of training, we conducted field tests, such as the VAMEVAL Test, the 5-Jump Test and the modified agility *t*-test.

By Vameval test, we noted a significant increase of MAS after the training period, in both groups after the training (group: *p* = 0.30, η^2^ = 0.13; time: *p* < 0.001, η^2^ = 0.81; interaction: *p* = 0.98, η^2^ = 0.000), but no difference between the two groups (Table 5).

The 5JT test was used to evaluate athletes’ lower limb explosive power and coordination level [40]. Group and time factors, as well as their interaction, had no significant effect on 5JT performance (group: *p* = 0.85, η^2^ = 0.004; time: *p* = 0.20, η^2^ = 0.19; interaction: *p* = 0.13, η^2^ = 0.26) and modified T test performance (group: *p* = 0.35, η^2^ = 0.11; time: *p* = 0.14, η^2^ = 0, 24; interaction: *p* = 0.41, η^2^ = 0.08).

On the other hand, the time of the modified agility *t*-test decreased significantly after 6-week of SSG training program (Table 5).

In particular, the time measured during the modified agility *t*-test was not significantly different between the two groups (HIIT and GSS), after medium intensity (80–90% of HRmax) or low intensity (<80% of HRmax) training programs. 

Thus, the results showed large in MAS, and medium in modified agility *t*-test changes following training intervention (Figure 5); only small in 5JT test (Figure 5).

## 4. Discussion

The aim of this study was to investigate the effects of high-intensity interval running (HIIT) and small-sided game training (SSG) programs on the physical, physiological, and psychological responses in young football players. After the 6-week training intervention, results showed no significant differences between the two groups with respect to the training load variables and the quality of total recovery; however, players indicated greater enjoyment and better well-being indices during SSG compared to HIIT.

Since the goal of high-level soccer training is to improve or maintain physical performance, it must be optimal based on an adequate balance between training volume and intensity and rest periods. When training is prolonged, excessive stress and possible inadequate recovery is suffered, by reaching states of overreaching or overtraining. As a result, sports performance decreases while psychological disturbances and the risk of injury increase [9].

Typically, training load was assessed via perceived exertion rating (RPE), heart rate (HR), and blood lactate concentration, and indeed, HR is most used to objectively monitor training intensity in many sports [45]. However, although HR is widely used and several studies have shown that it is a valid indicator of exercise intensity in football [46,47], it has some limitations. For example, as early as 1994, Bangsbo suggested that some factors relating to football training, as well as the discontinuous nature of sport, and the emotion, may cause to an overestimation of HR values of the real energy cost of exercise [48]. 

Conversely, other evidence demonstrate that HR underestimates the intensity of football drills with a high anaerobic component, even during short-duration SSGs involving few players [49]. Therefore, other measures of exercise intensity may provide a more appropriate assessment of exercise intensity during SSGs.

Blood lactate concentration [La] has also been used as an indicator of exercise intensity in football [29,32]. [La] was measured after each training session, just to evaluate differences in workload during HIIT and SSG. However, given the intermittent feature of football, blood [La] is a poor indicator of muscle lactate concentration during football match play [50]. In contrast to blood [La], the rating of Rated Perceived Exertion (RPE) is a facile and non-invasive method of monitoring exercise intensity [31]. Several studies have shown that RPE can be validly used to assess exercise intensity at a specific time [51] and as a global indicator of total session intensity (session RPE) [52,53]. RPE, due its psychobiological foundations [31], is a valid indicator of overall perception of effort for intermittent aerobic football-specific exercises (including SSGs) training [52,53].

In the present study, using RPE we assessed that training load progressively increased, in both SSG and HIIT training groups, reaching its maximum value in the 4th session, in agreement with results previously obtained by measuring blood [La] [24,32]. 

As training duration and intensity increased, we also observed an increase in training monotony and strain, an acceptable trend, for example, in a competitive period [33]. In this regard, monotony, represents the load variation within the week while, strain, represents the overall stress produced by the load over the week [33]. The results establish that the monotony value is about 1, in the 3th and 4th week, suggesting a balance between training and recovery during both training programs thus indicating that players do not reach the overcoming state [33,54]. Low values of monotony and strain correspond to a good balance between training and recovery, a condition strongly associated with the performance, training load and recovery of athletes [54,55]. 

Stressful factors such as training and competitive loads affect the performance and the general state of well -being of athletes [56]. Appropriate self-reporting questionnaires related to fatigue, stress level, DOMS and quality/sleep disorders [25] can measure the general well-being of the athlete. There is therefore the possibility of actually evaluating the state of well-being and the load perceived by providing the possibility of integrating different stimuli types [57] and to evaluate the difference between the trainings.

In general, well-being ratings are made to assess the impact of the congested period of training and matches [56,58], Ramadan fasting [59], and match-induced fatigue [60]. However, to the knowledge of the authors, no previous studies have yet compared these effects following two different training sessions, in elite young soccer players. Here, even if parallel to the training load, the subjective evaluations of the Hooper index reached their maximum values in the 4th week; however, low accumulated values and variability of well-being indicators were observed, indicating that the training was well tolerated by the players of both groups. However, when the relationships between load and well-being parameters are analysed in order to validate the Hooper index, no significant correlations are seen between the well-being and the RPE indicators during submaximal effort in footballers aged 17–19. Conversely, other studies revealed that perceived sleep, stress, fatigue and DOMS are related to professionally perceived daily load [56,61].

The comparison of the mood states (POMS) between the subjects engaged with SSG and with HIIT gives important information. The evaluation of the psychological state of the participants during the training and, in general, the physical activity can be obtained through POMS [62]. The present study found that the HIIT method caused a significant decrease in vigour and increase in total mood disturbance, unlike the SSG method which did not cause significant changes in the different POMS scores.

These results are consistent with research examining the relationship between exercise modality and mood in different sports disciplines [25,63,64]. Indeed, research has shown that vigour following intense training programs causes mood disturbances and reduced energy and commitment [65]. Thus, HIIT training program may induce negative feelings in players, unlike the SSG method [9,14]. It has been suggested that reduced vigour and increased TMD during HIIT are associated with lack of physical motivation and enjoyment associated with unpleasant psychological responses [13,15]. In SSG training, players are motivated by the technical and tactical aspects of the opposition [7,10,12], suggesting that the presence of the ball keeps the mood in the footballers. Also, the affective responses to training are an essential feature of a good performance [41]. Investigations that have utilized the PACES have evaluated enjoyment as a determinant of training sessions and related motivation [9,17,43], participation, and engagement in training [15]. The enjoyment and mood, in addition to focus and engagement, can involve adherence to training [66].

In the present study, we observed that PACES scores measured immediately after each training session were significantly higher in the SSG group than in the HIIT group, confirming previously studies performed in male soccer player [13,29]. In fact, exercise enjoyment and psychological state are consistently positively related [9,14,67,68,69]. Fundamental differences between SSG and HIIT able to exercise a strong influence on enjoyment reside in the use of the ball and in the presence of the opponents who effectively recreate the competitive atmosphere typically associated with the soccer competitions. Consequently, it is possible to correlate the aforementioned differences to the improvement of the level of play level and to the high level of enjoyment observed with SSG [68,70].

Generally speaking, professional soccer players are constantly exposed to various external stimuli during training in order to prepare them for the high intensity demands of a match [71,72]. These stimuli can induce changes associated with the execution of soccer technical skills and physical performance [73]. Accordingly, soccer coaches and practitioners usually monitor the current physical fitness status of their teams, and collect detailed information regarding each athlete [73,74] by periodic field tests that can assess player’s physical fitness status. Usually, it is preferred to conduct tests such as the VAMEVAL, but also the 5-Jump Test and the modified agility *t*-test are normally used in periodic assessments for analysing the cardiorespiratory, speed, and power capacities of soccer players [75]. The VAMEVAL test can measure maximum aerobic speed (MAS), allowing coaches to monitor progress and to calibrate the intensity of the training program [76]. Here, the increase in the MAS of both groups could indicate that both training protocols are effective in improving cardiorespiratory fitness in players. This suggests that rapid improvement in MAS is associated with both enhanced central functions, such as increased systolic stroke [77], as well as improved mitochondrial enzyme activity [10]. This finding is in line with the work of Hill-Haas et al. [10] who reported that the SSG training method is effective in improving the Intermittent Recovery Yo-Yo Test performance and in achieving heart rates (HR) similar to those found during HIIT training in soccer [78]. As examined in other studies in comparison to all other interventions, SSG excluded, HIIT induces a large mean positive effect on maximal running performance [22] and a small positive effect on maximal running performance when compared to SSG determined by incremental execution tests [13].

To address the need for a more specific agility test with change of direction speed, we have proposed a modified version of the classic agility *t*-test and, to our knowledge, no studies to date have compared the results of the modified agility *t*-test measured during HIIT and SGG training methods. Indeed, small differences between SSG and HIIT groups were observed during the modified agility *t*-test and 5JT test performances. However, a comparison between the impact of HIIT on repeated sprinting ability and that obtained with SSG was reported in one study involving a 12 × 20 m test [22] which produced a trivial negative effect. In a study on change-of-direction performance [79], HIIT had a large positive effect compared to SSG; this study contrasts with another in which greater improvement was observed with SSG [16], a discrepancy that cannot currently be explained. The effects on repeated sprint ability and change of direction performance both require further scientific evaluation.

Muscle power, measured using 5JT, did not change after 6 weeks of training with neither HIIT nor SSG, suggesting that both forms of training induce similar neuromuscular adaptations. These results are not in agreement with previous studies that have reported muscle fatigue and decreased muscle power with repeated bouts of high-intensity exertion, increased blood lactate concentrations, and limited recovery between attacks [80]. In addition, other studies have also indicated that 3 vs. 3 SSG produced significant alterations in muscle strength and balance after fatigue [81]. Regarding the comparison between HIIT and SGG previous studies report that HIIT had a trivial positive effect on jumping performance and small positive effects on the countermovement jump [13] and on the drop jump [79], as well as a small negative effect on the countermovement jump compared to SGG [79].

To summarize, our results show that the effects of SSG and HIIT are equally beneficial on the variables relating to the soccer-specific performance and to the endurance of the players, but they really have little impact on neuromuscular performance.

Otherwise, in youth female soccer players, both SSG and HIIT were shown effective for improving vertical and horizontal jumping ability, change-of-direction, and aerobic capacity [82]. Recently, in a study conducted on female soccer players from a Spanish professional club, SSG achieved improvements mainly in the variables related to repetition of effort and the ability to maintain acceleration and deceleration [83].

The present study has its limitations, and one of the main ones is related to the small sample size, which makes any generalization difficult. Therefore, conclusions should be interpreted carefully. However, considering that the sample is made up of young professional players, is almost impossible to have larger homogeneous participants. Another limitation of the study may be related to the variation in environmental conditions that occurred during the intervention period, and possibly different results can be found, in different phases of the season. Future studies should analyse longer period to make better generalizations. However, considering the conflicting evidence on the relationships between the dose of internal and external loads and changes in physical fitness, the present study revealed some interesting results. As practical implications, this study suggests that although field tests may show relationships between them, they are not necessarily expected to have similar dose-response relationships with internal and external training loads and with the same psychological and mood adjustments.

## 5. Conclusions

This study compared the physical, physiological and psychological responses obtained with HIIT and SSG training programs in elite young footballers and was conducted in a real-life training environment with competitive players providing important practical implications. And therefore, SSG can be considered a football-specific training that elicits physiological responses and neuromuscular adjustments and offers greater perceived enjoyment than what can be achieved with HIIT. Indeed, HIIT negatively affects the mood of the soccer players. For this reason, coaches of competitive youth soccer players should consider prioritizing the implementation of carefully designed SSGs over HIIT in their training programs.

Finally, further future investigations are needed to be done during other periods of the sporting season in order to compare these two training modalities, also using different parameters for SSG (i.e., the presence of goalkeepers, the duration of each bout, the encouragement of the coach, the size of the playing field, a different number of players, various periods of recovery and, finally, also different game rules). It will also be important in the future study to include players with different ages and different levels in order to extend the applicability of these results.

These results have a double use, the first concerns the coaches who can set up more performing training programs and the second concerns the young players who could improve their sports performances.

## Figures and Tables

**Figure 1 ijerph-19-13807-f001:**
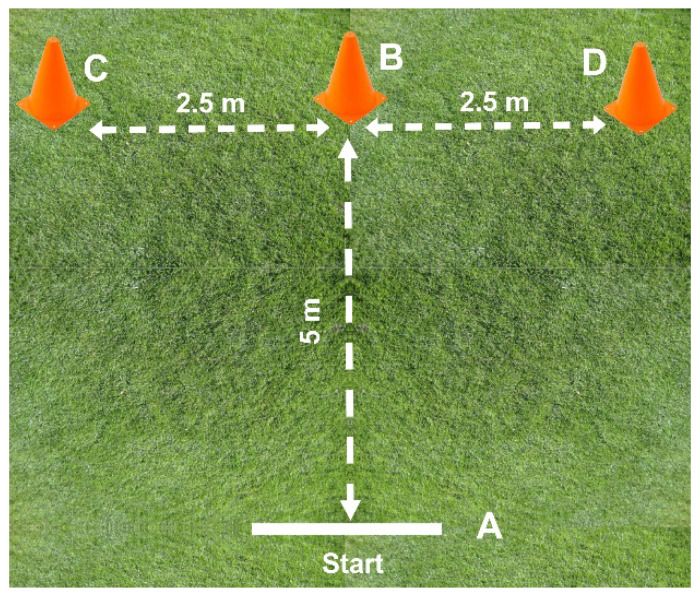
Track for the modified agility *t*-test.

**Figure 2 ijerph-19-13807-f002:**
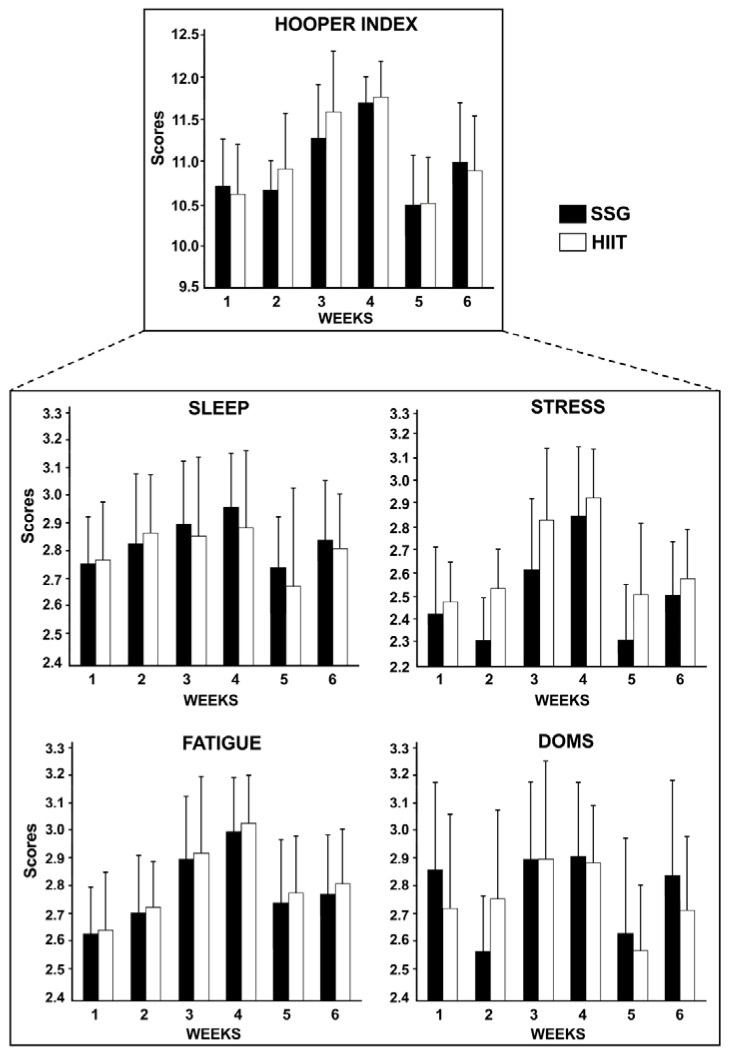
Hooper index which includes four subjective assessments: sleep, stress, delayed-onset muscle soreness (DOMS), and fatigue, measured during all the weeks of high intensity interval (HIIT) or small-sided games (SSG) training.

**Figure 3 ijerph-19-13807-f003:**
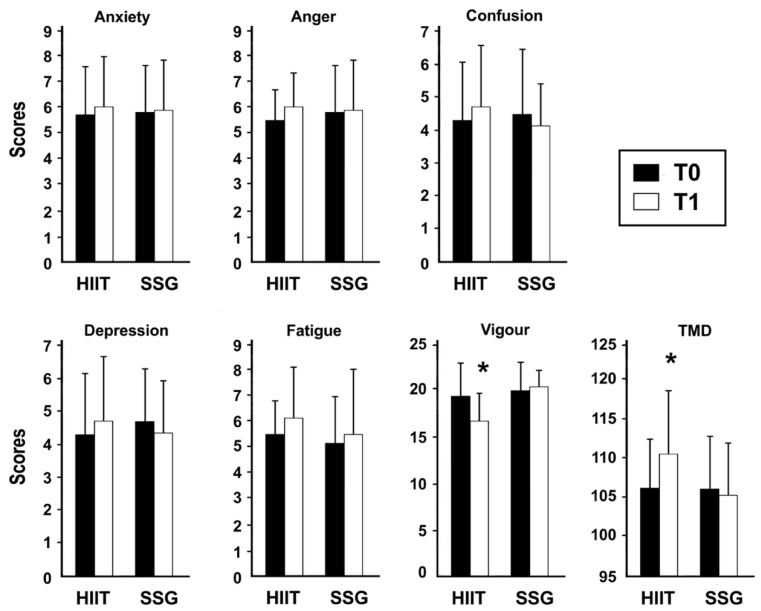
POMS scores for high intensity interval training (HIIT) and small-sided games (SSG) measured before and after the 6-week training program. TMD: total mood disorder. * Significant difference between before (T0) and after (T1) training for each group.

**Figure 4 ijerph-19-13807-f004:**
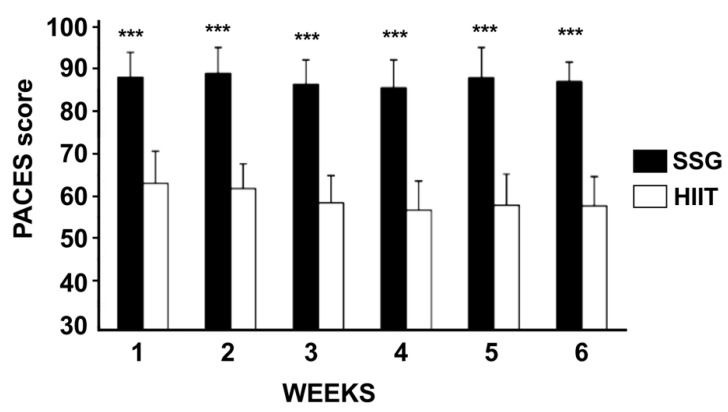
Physical Activity Enjoyment Score (PACES) measured in each week of high intensity interval (HIIT) or small-sided games (SSG) training. *** *p* < 0.01.

**Figure 5 ijerph-19-13807-f005:**
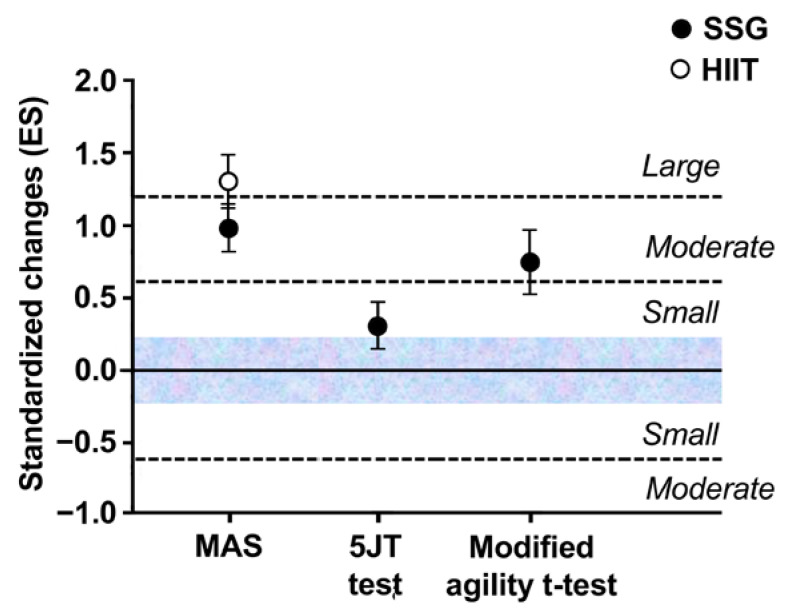
Relationships and sensitivities of Maximal aerobic speed (MAS), 5JT test and modified agility *t*-test to six-week high intensity interval (HIIT) or small-sided games (SSG) training program.

**Table 1 ijerph-19-13807-t001:** The characteristics of the subjects in HIIT and SSG group (means ± SD).

	HIIT	SSG	
Variables	Mean ± SD	Mean ± SD	*p*
Age (years)	16.8 ± 0.5	16.6 ± 0.6	0.7
Height (cm)	178.7 ± 8.6	178.58 ± 8.0	0.9
Body mass (Kg)	69.5 ± 6.5	68.7 ± 5.7	0.7
% Fat	11.1 ± 1.6	11.3 ± 2.0	0.9

*p* significance by independent *t*-test.

**Table 2 ijerph-19-13807-t002:** Description of 6-weeks training program.

Week	n° of Sessions	HIIT	SSG
1–6	12	25 min (4 × 4 min of 15″/15″ at 110% MAS, 3 min rest)	25 min (4 × 4 min, at maximum effort 3 min rest)
7	Post-intervention testing		

Min, minute; MAS, maximum aerobic speed.

**Table 4 ijerph-19-13807-t004:** POMS scores measured before and after high intensity interval training (HIIT) or small-sided games (SSG).

Variables	Main Effect of the Condition		Main Effect of the Times		Interaction Effect	
	F	η^2^	F	η^2^	F	η^2^
**Anxiety**	0.00	0.00	0.64	0.07	0.23	0.03
**Anger**	0.10	0.01	0.28	0.22	0.50	0.06
**Confusion**	0.64	0.07	0.000	0.000	1	0.11
**Depression**	0.00	0.00	0.000	0.000	2.66	0.25
**Fatigue**	2.76	0.25	2.40	0.23	0.44	0.05
**Vigour**	18.18 **	0.69 **	3.42	0.30	18.28 **	0.70 **
**TMD**	10.32 *	0.56 *	4.33 *	0.35 *	7.62 *	0.48 *

TMD: total mood disturbance. Significant effects are shown, with * *p* <0.05, ** *p* <0.01.

**Table 5 ijerph-19-13807-t005:** Mean (± SD) values for physical performance variables for high intensity interval (HIIT) and small-sided games (SSG) training, measured before (T0) and after (T1) the six-week training program.

Variables	Group	T0	T1	|d|	η^2^	Evaluation of the Differences
**MAS (Km·h^−1^)**	HIIT	17.07 ± 0.57	17.63 ± 0.36	0.56 ***	1.26	Large
	SSG	16.83 ± 0.72	17.38 ± 0.49	0.55 ***	0.95	Large
**5-jump test (m)**	HIIT	11.88 ± 0.78	11.89 ± 0.66	0.01	0.01	Trivial
	SSG	11.73 ± 0.72	11.93 ± 0.55	0.20	0.33	Small
**Modified** **agility *t*-test (s)**	HIIT	5.92 ± 0.40	5.88 ± 0.22	0.04	0.13	Trivial
	SSG	5.87 ± 0.26	5.72 ± 0.17	0.15 *	0.75	Moderate

MAS: Maximal aerobic speed; η^2^: effect size. * *p* < 0.05; *** *p* < 0.01. The extent of the differences between T0 and T1 was assessed by the Hopkins threshold [43].

## Data Availability

The raw data supporting the conclusions of this article will be made available by the authors without undue reservation.

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
