# Peer review of "Comparison of the Physical, Physiological, and Psychological Responses of the High-Intensity Interval (HIIT) and Small-Sided Games (SSG) Training Programs in Young Elite Soccer Players"

_ijerph, 2022, doi:10.3390/ijerph192113807_

Round 1

Reviewer 1 Report

I suggest the authors make a revision or adjustment as follows:

Line 75. In Table 1, I suggest you divide the subjects into HIIT and SSG groups and make a comparison with an independent t-test on age, height, body mass, and % fat to demonstrate there's no significant difference between HIIT and SSG groups at the baseline. Because in your paper, you only referred to a randomized assignment, but the readers do not know about the result after the randomized assignment at the baseline.

Line 99. A spelling mistake, the perceived exertion (RPE) should be Rated Perceived Exertion (RPE), revise it.

Line 133. The same mistake as above. Change rating of perceived exertion (RPE) into Rated Perceived Exertion (RPE).

Line 175. I recommend using a subsection to explain those tests. For example, after 2.10. Physical Performance use 2.10.1 “5-Jump Test” and move the related contents in the paragraph to the next line, etc.

Line 177. Delete a space before previously and delete a comma before [40].

Line 204. Insert a space before 42.

Line 222-224. I suggest you shorten caption 3.1 into a phrase in order to maintain the balance with captions 3.2 and 3.3.

Line 228. Add right-half parentheses after Table 2.

Line 446. A spelling mistake. Change SGG into SSG.

Line 473-477. I suggest you move this paragraph to the last paragraph of the discussion section after limitations. It means, apart from the limitations in your study, what aspects should be improved in future research?

Finally, from your study, we know that SSG can be considered an effective, more motivating, and fun soccer-specific training that elicits physiological responses and neuromuscular adjustments in young soccer players. However, we don’t know how to use the SSG program in the daily practice among young soccer players. Could you supplement the related information in discussion section?

Author Response

I would like to thank you for the evaluation on our manuscript.

We have made the suggested changes to our manuscript.

Line 75. In Table 1, I suggest you divide the subjects into HIIT and SSG groups and make a comparison with an independent t-test on age, height, body mass, and % fat to demonstrate there's no significant difference between HIIT and SSG groups at the baseline. Because in your paper, you only referred to a randomized assignment, but the readers do not know about the result after the randomized assignment at the baseline.

We have done new Table 1, and we added:

Consistently with a random assignment of two groups, no significant differences between groups were found in mean values for age, height, body mass, and % fat, at baseline (p > 0.05 by independent t-test).

Line 99. A spelling mistake, the perceived exertion (RPE) should be Rated Perceived Exertion (RPE), revise it.

We have done 

Line 133. The same mistake as above. Change rating of perceived exertion (RPE) into Rated Perceived Exertion (RPE).

We have done 

Line 175. I recommend using a subsection to explain those tests. For example, after 2.10. Physical Performance use 2.10.1 “5-Jump Test” and move the related contents in the paragraph to the next line, etc.

We have done 

Line 177. Delete a space before previously and delete a comma before [40].

We have done

Line 204. Insert a space before 42.

We have done

Reviewer 2 Report

Very well conducted study and very well written.  Line  362 avoid sweeping statements - In general.  

More about the stakeholders in your conclusion.  

Again - well done. 

Author Response

Dear

I would like to thank you for the evaluation on our manuscript.

Line 362:

We have replaced the sentence with:

The analysis of perceived load and wellness status could have the potential to integrate different types of stimuli ..”

More about the stakeholders in your conclusion.  

We added:

These results have a double use, the first concerns the coaches who can set up more performing training programs and the second concerns the young players who could improve their sports performances.

Reviewer 3 Report

The paper is well-written and reports a well-conducted study. I applaud the authors' efforts and encourage their pursuit for publication. I do think, however, that they should target another journal, which has a particular focus on sports and exercise. The scope, focusing on a particular form of soccer practice exercise for young players, is far to limited to make a significant contribution to public health and the readership of this journal.
It is unclear how the findings can expand beyond the context of soccer practice for young players. As the authors note, the physiological effects have already been investigated and are basically replicated in the present study. The motivational and psychological effects, on the other hand, seem rather intuitive: young soccer players react more favorably to exercise that relates to soccer.
I do think, however, that this would be of greater interest to a sports and exercise journal readership. 

Author Response

The paper is well-written and reports a well-conducted study. I applaud the authors' efforts and encourage their pursuit for publication. I do think, however, that they should target another journal, which has a particular focus on sports and exercise. The scope, focusing on a particular form of soccer practice exercise for young players, is far to limited to make a significant contribution to public health and the readership of this journal.

It is unclear how the findings can expand beyond the context of soccer practice for young players. As the authors note, the physiological effects have already been investigated and are basically replicated in the present study. The motivational and psychological effects, on the other hand, seem rather intuitive: young soccer players react more favorably to exercise that relates to soccer.

I do think, however, that this would be of greater interest to a sports and exercise journal readership.

Thanks for your appreciation

Regarding your note, we believe that this study also has a general interest regarding the sports activities of adolescents practicing soccer as it specifically concerns the effects of certain types of training on the psychological mood, we can assume that it concerns public health. 

Reviewer 4 Report

This is a very good study, in a field of great interest.The paper is well written and the references seem update. However, I have a few mentions to do:

 Introduction:

- Lines 61-66

What are the concrete data that emerge from the studies mentioned by you (22,23) regarding the use of HIIT and SSG methods? Practically, the idea of initiating the research started from these and I think that they should be presented briefly, so that the reader knows them.

- Lines 66-68

I think you meant to state the purpose of the paper here, but it's quite unclear. Why did you want to compare the physical, physiological and psychological responses of the two training programs? What did you want to prove with this comparison – the working hypothesis? You need to be a little bit more clear in stating the purpose of this research and specify what is the working hypothesis you started from, especially since in 2.1 you mention the "null hypothesis" (line 85).

-         Please formulate some general conclusions that emerge from the studies mentioned in this chapter.

2.     Materials and Methods

2.1.          Participants

-         In lines 78-79 you say that the subjects were divided into two groups according to their playing position, and in lines 81-82 you say that the distribution was done randomly (by flipping a coin). The idea is slightly confusing. Please explain better how the two groups were assigned. It would also be better to specify whether the profile of the two groups (physical, respectively physiological) was similar at the beginning of the research.

2.2. Study design

-         When and during what period (6 weeks starting on ..... and ending on ........) was this research conducted? In what preparation period of the year (preparatory stage but which? general, specific, multilateral, etc., the results also depend on this!)

-         Lines 92-93 - extension... I must understand that after the end of the main training or included in it, but with the increase in volume/duration of time? I think this aspect should be mentioned.

-         Lines 93-94 - "Before and after the interventions...", that is, twice a week or at the beginning and at the end of the study? If twice a week, should it be specified what is the interval between the two training sessions?

-         Line 96 – I don't understand.... daily, meaning from Monday to Saturday or after every training session? 

2.3. Small-Sided Games (SSG)

-         Lines 116-118 – did the two coaches only offer new balls or did they also contribute with recommendations, encouragements, observations addressed to the players? I think they can influence the psychological side of the players.

-         If all SSG were played without a goalkeeper, how were the other positions organized in the game? 

2.5. Measurements of Exercise Intensity

-         Lines 140-141 – the phrase is repeated. Please delete it. 

2.7 Total quality of recovery (TQR)

-         the first daily training session - does it refer to the HIIT and SSG training sessions or the 5 training sessions / week?

2.9. The Profile of Mood States (POMS)

-         reference 38 is a book. Please mention in the references which pages you are referring to! 

2.10. Physical Performance

-         In order to follow a logical sequence, I think this paragraph should be moved after 2.5.

-         Line 201 – what breaks were there between the 3 attempts? I think it should be specified! Were all players tested at the same time? If not, what were they doing during this time?

References:

I recommend that the references be reviewed because not all of them, fully meet the requirements. No DOI was included for all references where it exists.

Author Response

I would like to thank you for the evaluation on our manuscript and the suggested changes to our manuscript.

 Introduction:

- Lines 61-66

What are the concrete data that emerge from the studies mentioned by you (22,23) regarding the use of HIIT and SSG methods? Practically, the idea of initiating the research started from these and I think that they should be presented briefly, so that the reader knows them.

We have added:

Interestingly, to our knowledge, only a few studies [22,23] have examined the effects of HIIT and SSG methods on the physical fitness and the level of soccer-specific technical skills in young soccer players (U-16) and none from a motivational point of view.

In particular, the results of a meta-analysis of the relevant scientific literature suggested that HIIT or SSG, also in young soccer, produced similar positive effects in the development of aerobic capacity, power, speed, endurance at speed and football specific performance, but little influence on neuromuscular performance [22]. Additionally, some studies suggested that the competition that occurs in the SSG motivates and facilitates players to reach levels of high-intensity exercise [22]. In fact, many soccer players do not reach their performance potential not just because they lack during training the simulations necessary for the harmonious development of their aerobic capacity, but also for the lack of state of positive and motivated mood.

- Lines 66-68

I think you meant to state the purpose of the paper here, but it's quite unclear. Why did you want to compare the physical, physiological and psychological responses of the two training programs? What did you want to prove with this comparison – the working hypothesis? You need to be a little bit clearer in stating the purpose of this research and specify what is the working hypothesis you started from, especially since in 2.1 you mention the "null hypothesis" (line 85).

We added:

Therefore, it would be important to determine which of the two training programs, HIIT or SSG, produces the strongest motivational responses in young players, taking into account that structuring situations that are intrinsically interesting, challenging and amusing could develop and enhance intrinsic motivation.

Thus, the present study attempted to compare not only the physical and the physiological adaptations but also the psychological responses of the two training programs in young soccer players.

Please formulate some general conclusions that emerge from the studies mentioned in this chapter.

We done

  1. Materials and Methods

2.1.          Participants

-         In lines 78-79 you say that the subjects were divided into two groups according to their playing position, and in lines 81-82 you say that the distribution was done randomly (by flipping a coin). The idea is slightly confusing. Please explain better how the two groups were assigned. It would also be better to specify whether the profile of the two groups (physical, respectively physiological) was similar at the beginning of the research.

Regarding the assignment to the HIIT group (n = 12) or to the SSG group (n = 12) first, all soccer players were divided into two groups (n=12 each) based on their playing position (central defender, full-back, attacking midfielder, attacking midfielder and forward). Then, they were randomly assigned to each group, SSG or HIIT, by flipping a coin.

And we added:

Consistently with a random assignment of two groups, no significant differences between groups were found in mean values for age, height, body mass, and % fat, at baseline (p > 0.05 by independent t-test).

2.2. Study design

-         When and during what period (6 weeks starting on ..... and ending on ........) was this research conducted? In what preparation period of the year (preparatory stage but which? general, specific, multilateral, etc., the results also depend on this!)

We added:

The study was conducted during the 2021-2022 football season, in Tunisia. It started after the pre-season preparation, on October 19th and ended on December 29th, for a duration of 6 weeks.

-         Lines 92-93 - extension... I must understand that after the end of the main training or included in it, but with the increase in volume/duration of time? I think this aspect should be mentioned.

The training interventions were administered as an extension of the regular training; two sessions / week of HIIT or SSG training were performed after a standardized warm-up, for 25 minutes and were strictly controlled as will be described later (in 2.3. Small-Sided Games (SSG)).

-         Lines 93-94 - "Before and after the interventions...", that is, twice a week or at the beginning and at the end of the study? If twice a week, should it be specified what is the interval between the two training sessions?

To clarify the description " Before and after the interventions...", it has been replaced with "Before and after 6 weeks of training"

-         Line 96 – I don't understand.... daily, meaning from Monday to Saturday or after every training session? 

We made it clearer with the with the addition of following sentences:

Before and after 6 weeks of training, all participants completed the evaluations of anthropometric parameters, the VAMEVAL test, the 5 jumps test (5JT) and the Hooper Index and POMS questionnaires. 

For the follow-up we used a training diary in which the participants recorded twice a week (when they performed the HIIT or SSG training sessions), the indices of well-being (the quality of sleep, stress, the level of fatigue and the delayed-onset muscular soreness (DOMS)) [25], the quality of total recovery (TQR) [26], (the working time (in min), the Rated Perceived Exertion (RPE) [27].

In relation to the elements collected (duration of work (in min) and RPE), data processing was carried out to determine the training load, monotony and stress on a daily, weekly and periodic basis.

2.3. Small-Sided Games (SSG)

-         Lines 116-118 – did the two coaches only offer new balls or did they also contribute with recommendations, encouragements, observations addressed to the players? I think they can influence the psychological side of the players.

We added:

As coaches can psychologically influence the players, they constantly verbally encourage players from both groups, also giving instructions to improve the training /game using football specific terminology

-      If all SSG were played without a goalkeeper, how were the other positions organized in the game? 

We added:

During the training sessions of the SSG, the players freely occupied the entire space of the pitch.

2.5. Measurements of Exercise Intensity

-         Lines 140-141 – the phrase is repeated. Please delete it. 

We have done.

2.7 Total quality of recovery (TQR)

-         the first daily training session - does it refer to the HIIT and SSG training sessions or the 5 training sessions / week?

TQR was recorded by each player, 15 minutes before each HIIT or SSG session (two sessions / week).

2.9. The Profile of Mood States (POMS)

-         reference 38 is a book. Please mention in the references which pages you are referring to! 

We done

2.10. Physical Performance

-         In order to follow a logical sequence, I think this paragraph should be moved after 2.5.

We believe the paragraph is better off in its original position as these results are presented later.

-         Line 201 – what breaks were there between the 3 attempts? I think it should be specified! Were all players tested at the same time? If not, what were they doing during this time?

Tests were performed before the study and after the 6-week period. The tests were distributed on two consecutive days: on the first day the players performed the Modified agility t-test and after 10 minutes of rest, they performed the 5-Jump Test (5JT); on the second day the players performed the Vameval test. To minimize the effects of external variables prior to the test, participants observed a 48-hour rest period, diet and sleep period similar for all.

Before the test, participants completed a 10-minute warm-up period, including sprinting, lateral shifting, jumping, and dynamic stretching. Every 10 minutes, 4 players were called by the coach to perform the tests, while the other players were waiting for their turn near the field. Each player performed 3 trials alternating with 3 minutes of passive recovery, to ensure adequate recovery [42].

References:

I recommend that the references be reviewed because not all of them, fully meet the requirements. No DOI was included for all references where it exists.

We done.

Reviewer 5 Report

Dear authors,

Thank you very much for your contribution to your research. Your research is valuable in terms of its scope and content, I think that your research is suitable for publication in ijerph journal after necessary adjustments that I would like you to make only a few corrections.

-Please edit your title, including HIIT and Small side games in your title will increase the legibility of your research. Because you didn't make use of them either in the title or the keywords.

Abstract

Please divide your research into sections (Background and objectives, materials and method, results, conclusion) and add a short background in front of your purpose statement.

Introduction

Please add a sentence containing your main hypothesis at the end of this section.

materials and methods

-I recommend that you show the HIIT and SSG workouts in detail in a tabular form.

-Please indicate in what order you did the tests in the study design, and how was the rest between the tests? As you know, not all tests can be done on the same day and in the same order. The way you follow the tests is important to support the article findings.

Results

I think that this section is done with adequate and correct presentation. There is no need for any corrections.

Discussion

In your discussion, you have made reference to many studies related to your subject, but I will add a few studies that were directly related to your research and published in 2022, and I suggest you take advantage of them. As a matter of fact, the information you will receive from here will contribute to the evaluation of your results by gender groups.

Nayiroglu, S., Yilmaz, A.K., Silva, A.F. et al. Effects of small-sided games and running-based high-intensity interval training on body composition and physical fitness in under-19 female soccer players. BMC Sports Sci Med Rehabil 14, 119 (2022). https://doi.org/10.1186/s13102-022-00516-z

Nevado-Garrosa, F., Torreblanca-Martinez, V., Paredes-HernÁndez, V., & Balsalobre-FernÁndez, C. (2021). Effects of an eccentric overload and small-side games training in match accelerations and decelerations performance in female under-23 soccer players. The Journal of Sports Medicine and Physical Fitness, 61(3), 365-371.

Best Regards

Author Response

Thank you very much for your appreciation

 According to the indications of the refery, we change the title to:

Comparison of the physical, physiological, and psychological responses of the high-intensity interval (HIIT) and small-sided games (SSG) training programs in young elite soccer players

Abstract

Please divide your research into sections (Background and objectives, materials and method, results, conclusion) and add a short background in front of your purpose statement.

Unfortunately, the formatting suggested by the Journal IJERPH  does not allow us to exceed 200 words in the abstract. So we are forced not to change it

Introduction

Please add a sentence containing your main hypothesis at the end of this section.

We added:

 Therefore, it would be important to determine which of the two training programs, HIIT or SSG, produces the strongest motivational responses in young players, taking into account that structuring situations that are intrinsically interesting, challenging and amusing could develop and enhance intrinsic motivation.

Thus, the present study attempted to compare not only the physical and the physiological adaptations but also the psychological responses of the two training programs in young soccer players.

Materials and methods

-I recommend that you show the HIIT and SSG workouts in detail in a tabular form.

We added:

Table . Description of 6-weeks training program

Week

N° of Sessions

HIIT

SSG

1-6

12

25min

(4 × 4 min of 15’’/15’’

at 110% MAS,

3 min rest)

25 min

(4 × 4 min,

at maximum effort

3 min rest)

7

Post-intervention testing

Min, minute; MAS, maximum aerobic speed

-Please indicate in what order you did the tests in the study design, and how was the rest between the tests? As you know, not all tests can be done on the same day and in the same order. The way you follow the tests is important to support the article findings.

We added:

Tests were performed before the study and after the 6-week period. The tests were distributed on two consecutive days: on the first day the players performed the Modified agility t-test and after 10 minutes of rest, they performed the 5-Jump Test (5JT); on the second day the players performed the Vameval test. To minimize the effects of external variables prior to the test, participants observed a 48-hour rest period, diet and sleep period similar for all. 

Discussion

In your discussion, you have made reference to many studies related to your subject, but I will add a few studies that were directly related to your research and published in 2022, and I suggest you take advantage of them. As a matter of fact, the information you will receive from here will contribute to the evaluation of your results by gender groups

We added:

Otherwise, in youth female soccer players, both SSG and HIIT effectively improved vertical and horizontal jumping ability, change-of-direction, and aerobic capacity [82]. Recently, in a study conducted on female soccer players from a Spanish professional club, SSG achieved improvements mainly in the variables related to repetition of effort and the ability to maintain acceleration and deceleration [83].

Round 2

Reviewer 5 Report

Dear Authors,

Congratulations. Thank you for your efforts.

Best Regards.

Author Response

Thank you very much for your guidence.